# Colonization and Infection of Indwelling Medical Devices by *Staphylococcus aureus* with an Emphasis on Orthopedic Implants

**DOI:** 10.3390/ijms23115958

**Published:** 2022-05-25

**Authors:** Giampiero Pietrocola, Davide Campoccia, Chiara Motta, Lucio Montanaro, Carla Renata Arciola, Pietro Speziale

**Affiliations:** 1Department of Molecular Medicine, Biochemistry Section, Viale Taramelli 3/b, 27100 Pavia, Italy; giampiero.pietrocola@unipv.it (G.P.); chiara.motta02@universitadipavia.it (C.M.); 2Laboratorio di Patologia delle Infezioni Associate all’Impianto, IRCCS Istituto Ortopedico Rizzoli, Via di Barbiano 1/10, 40136 Bologna, Italy; davide.campoccia@ior.it (D.C.); lucio.montanaro@unibo.it (L.M.); 3Department of Experimental, Diagnostic and Specialty Medicine, University of Bologna, Via San Giacomo 14, 40126 Bologna, Italy; 4Laboratorio di Immunoreumatologia e Rigenerazione Tissutale, IRCCS Istituto Ortopedico Rizzoli, Via di Barbiano 1/10, 40136 Bologna, Italy

**Keywords:** indwelling medical devices, *Staphylococcus aureus*, biofilm, infection

## Abstract

The use of indwelling medical devices has constantly increased in recent years and has revolutionized the quality of life of patients affected by different diseases. However, despite the improvement of hygiene conditions in hospitals, implant-associated infections remain a common and serious complication in prosthetic surgery, mainly in the orthopedic field, where infection often leads to implant failure. *Staphylococcus aureus* is the most common cause of biomaterial-centered infection. Upon binding to the medical devices, these bacteria proliferate and develop dense communities encased in a protective matrix called biofilm. Biofilm formation has been proposed as occurring in several stages—(1) attachment; (2) proliferation; (3) dispersal—and involves a variety of host and staphylococcal proteinaceous and non-proteinaceous factors. Moreover, biofilm formation is strictly regulated by several control systems. Biofilms enable staphylococci to avoid antimicrobial activity and host immune response and are a source of persistent bacteremia as well as of localized tissue destruction. While considerable information is available on staphylococcal biofilm formation on medical implants and important results have been achieved on the treatment of biofilms, preclinical and clinical applications need to be further investigated. Thus, the purpose of this review is to gather current studies about the mechanism of infection of indwelling medical devices by *S. aureus* with a special focus on the biochemical factors involved in biofilm formation and regulation. We also provide a summary of the current therapeutic strategies to combat biomaterial-associated infections and highlight the need to further explore biofilm physiology and conduct research for innovative anti-biofilm approaches.

## 1. Introduction

Advances in surgery and biomaterials science have provided us with a wide range of artificial body parts that can be used to replace/repair their natural counterparts [1,2,3,4,5]. The absolute number of implant-related infections has been documented as steadily growing for the aging population. This is particularly true in orthopedics as the prevalence of osteoarthritis, the principal cause of joint diseases and implant surgery, increases indefinitely with age. As recalcitrant to antibiotic therapy and surgical curettage, infection is a serious complication of implant orthopedic surgery, often leading to implant failure [6]. Moreover, it must be mentioned that each substitution of a failed prosthesis involves a much higher risk of relapse (up to 30%) than that for primary surgery. A number of medical procedures involve the insertion of a variety of catheters through the skin or orifices, and other clinical areas are fertile grounds for the application of biomaterials. Biomaterials may be natural or synthesized in the laboratory using a variety of chemical approaches depending on what they will be used for. Naturally derived biomaterials can be classified in at least two groups including protein-based (collagen, gelatin, silk) and polysaccharide-based biomaterials (cellulose, chitin/chitosan). Synthetic biomaterials are generally grouped into metals, ceramics, plastic, glass and polymers. Furthermore, significant research has investigated creating composites of these materials to combine their benefits [7]. Biomaterials can be functionally passive, such as those used for a heart valve, or may be bioactive with a more interactive functionality such as hydroxyapatite-coated hip implants [8].

On the one hand, biomedical implants have revolutionized medicine and opened new avenues in human health progress, but they have also increased the risk of infection of body tissues. In fact, while in the absence of a foreign implant, body contamination by an opportunistic pathogen is usually cleared by host immune defenses, the implantation of a medical device exposes the body to the risk of a permanent colonization and potential replacement [9,10].

Bacterial adhesion is the first step of biomaterial-associated infections. Staphylococci are the most frequent causes of device-associated infections, with *Staphylococcus aureus* and *Staphylococcus epidermidis* representing the leading species of contamination [11]. This successful action is due to the large variety of virulence factors expressed by these pathogens [12,13]. In many circumstances, indwelling medical devices are contaminated by bacteria during surgery. The source can be the skin or other colonized body sites of the patients or the health care workers. In an alternative route, the already implanted devices can be infected via hematogenous seeding. Moreover, immunocompromised patients, such as AIDS patients or premature newborns, are at increased risk of developing infections when they receive a medical implant.

The ability of *S. aureus* to promote the formation of a three-dimensional structure composed of the bacteria and extracellular biopolymers (polysaccharides and/or proteins) referred to as biofilm is central to the biomedical infection process [14]. Biofilm can be defined as a microbial community of bacteria that is attached to a substratum and embedded in a matrix that they have produced. A prerequisite for biofilm formation is the coating of medical devices with plasma proteins [6,14]. Bacterial cells in a biofilm exhibit a phenotype that is different in terms of growth rate and gene transcription with respect to that exhibited by planktonic bacteria. The physiological features and structure of bacteria in biofilm may explain the increased resistance to antimicrobial agents and less susceptibility to biological attacks by the immunological defenses of the host [14,15]. In continuity with our previous report [6], in this review we focus on the types of biomedical implants, with a special emphasis on the orthopedic materials, and update the role of adhesive plasma proteins in coating biomaterials and the function of *S. aureus* virulence factors in the colonization and infection of indwelling medical devices. We also discuss in more detail the structure and function of biofilm as an important means against immune evasion of the host. Finally, we illustrate the potential strategies to prevent and therapeutically treat biomaterial-associated infections by *S. aureus*. We believe that the holistic approach implemented in this study where microbiological, biochemical, and material science data were integrated may offer a useful cultural framework for microbiologists, experts in biochemistry and clinicians.

## 2. Most Used Medical Devices: Chemical Composition, Applications and Infection

Medical devices particularly prone to *S. aureus* infection include, in addition to prosthetic joints, central venous catheters, urinary catheters, endotracheal tubes, mechanical heart valves, pacemakers and contact lenses.

*Prosthetic joints.* A variety of materials have been used for orthopedic implants. High-molecular-weight polyethylene is the main polymer used for artificial joints, and it is widely used as an acetabular material component due to its superior biocompatibility, non-toxicity and high impact strength. Due to their good chemical stability, excellent resistance to corrosion and reliable long-term behavior, alumina and zirconia ceramics have been extensively used in total joint bearings [16]. Silicon carbide, classified as a non-oxide ceramic, has higher strength and hardness characteristics than those of alumina. Even better, stainless steel, Co-based alloys and titanium and its alloys, due to their good elasticity, superior biocompatibility and wear resistance, are the most widely used metallic materials in orthopedic replacement [17,18]. Polymethyl methacrylate (PMMA) is the well-known orthopedic fixation cement. It has been used extensively over the last 50 years to make stable joint implants (mainly hip or knee) or as a bone filler. The main function of an orthopedic biomaterial is to bear load, also providing structural integrity and stability to the human body. The nanocoating of metals and alloys are promising orthopedic materials for controlling protein adsorption and tissue integration at the bone-implant interface, improving implant functionalities, and strengthening mechanical properties of orthopedic implants. *S. aureus* and the coagulase-negative staphylococci are the major cause of prosthetic joint infection [19]. Hematogenous seeding or surgery interventions of arthroplasty, where the articular surface of a musculoskeletal joint is remodeled or replaced, can cause the infection of the orthopedic device by *S. aureus*, the development of bacterial macroscopical biofilm-like clumps in the synovial fluid present in the joints and consequent recalcitrance of infection to antibiotics [20].

*Mechanical heart valves*. Mechanical heart valves are made of materials such as titanium and carbon and are commonly used to replace aortic and mitral valves damaged with age, by certain diseases or congenital abnormalities. They usually consists of two leaflets and a metal ring surrounded by a ring of knitted fabric, which is sewn onto the heart in place of the original valve. Both *S. epidermidis* and *S. aureus* form biofilms on mechanical heart valves and the surrounding cardiovascular tissues, and in some cases this can result in serious diseases such as prosthetic valve endocarditis [21,22,23].

*Central venous catheters (CVCs).* CVCs are used to consign blood products, nutrients and medications, as well as facilitating dialysis treatment. Virtually all current central venous catheters (particularly long-term catheters) are made of silicone rubber or polyurethane. The internal jugular vein, common femoral vein and subclavian veins are the preferred sites for temporary central venous catheter placement, while for mid-term and long-term central venous access, the basilic and brachial veins are utilized for peripherally inserted central catheters. CVCs are predominantly infected by *S. epidermidis* and *S. aureus* [24]. In the case of *S. aureus* infections, typical clinical signs include purulence, erythema and tenderness [25].

*Urinary catheters*. Urinary catheters are tubular silicone or latex devices used to empty the bladder and collect urine in a drainage bag. Biofilms can readily develop on the inner or outer surfaces of urinary catheters upon insertion, making it difficult the prevention of bacterial colonization by using mere hygienic procedures [26]. The dominant organisms isolated from urinary catheters are *S. epidermidis*, *S. aureus*, *Enterococcus faecalis*, and *Escherichia coli* [27,28]. 

*Endotracheal tube*. An *endotracheal tube* is a flexible polyvinyl chloride tube that is placed between the vocal cords through the trachea. The endotracheal tube is then connected to a ventilator, which delivers oxygen to the lungs. Endotracheal tubes directly link the outside environment and the lungs, making them susceptible to bacterial infection. For this reason, biofilms can rapidly develop on endotracheal tubes [29]. *S. aureus* is the most frequent pathogen infecting ventilator-associated pneumonia, next to *Pseudomonas aeruginosa* [30].

*Cerebrospinal shunts*. A shunt is a hollow tube surgically placed in the brain (or occasionally in the spine) to allow the drainage of cerebrospinal fluid and redirect it to another location in the body. Cerebrospinal shunt infections remain the most significant complication in the treatment of hydrocephalus and significantly contribute to excess morbidity and mortality. Skin flora organisms such as *S. epidermidis*, *S. aureus*, and *Propionibacterium acnes* are the most common infecting organisms, and the clustering of approximately 70% of infections within the two-month postoperative period strongly suggests that colonization during shunt placement is an important initiating event [31].

*Contact lenses*. Contact lenses (CLs) can be classified as hard and soft. Conventional hard CLs are made of fairly stiff plastic (polymethyl methacrylate or PMMA) and are rigid and impermeable to oxygen [32,33]. Hard permeable CLs are made of silicone-containing compounds that allow oxygen to pass through the lens material to the cornea [34]. The first type of soft CLs are hydrogels made of 2-hydroxyethylmethacrylate (HEMA) cross-linked to ethylene glycol dimethacrylate and contain up to 38% water [35]. Silicone hydrogel contact lenses are more advanced soft lenses that allow more oxygen to pass through the lens to the cornea than regular hydrogel contacts and may be up to 70% water [36]. During contact lens wear, proteins in the tear film deposit on the CLs and promote microbial adherence. Tears contain more than a thousand proteins that exert various functions, such antimicrobial defense, lubrification and would healing. Recently, Osei et al. reported that the adsorption of the glycoprotein Gp340, a normal component of tears, on two common contact lens materials promotes the adhesion of *S. aureus* and *Pseudomonas aeruginosa*. This study also identified staphylococcal surface protein SraP as a receptor for Gp340 [37]. Since *S. aureus* is among the most common causes of ocular infections this information could have potential therapeutic implications. Table 1 summarizes the major biomaterials used in medicine and their properties.

## 3. Staphylococcal Biofilm Development on the Surface of Biomaterials

As reported above, biofilms are communities of sessile bacteria tightly adhering to biomaterial surfaces, encased in a matrix of extracellular polymeric substances [6,38]. In many bacteria including *S. aureus* biofilm formation involves several stages: initial attachment; accumulation and maturation; dispersal (Figure 1). In all these stages proteinaceous and non-proteinaceous compounds are expressed and their expression is tightly controlled [39,40].

### 3.1. Attachment of S. aureus Cells to Plasma Protein-Coated Biomaterials

Staphylococcal biofilm formation on medical devices begins with the attachment of staphylococcal cells to the abiotic surface followed by adhesion to the plasma protein-coated surface of the biomaterials. Interactions between staphylococci and the biomaterial surface are not specific and are driven by hydrophobic, electrostatic and Van der Waals forces (Figure 1). The hydrophobic character of a few staphylococcal surface components, such as wall teichoic acids and the surface proteins Atl from *S. aureus* and *S. epidermidis*, also contribute to the formation of hydrophobic interactions between bacteria and the material surface [41,42,43]. However, in vivo direct interaction of *S. aureus* cells with the abiotic implant surface is not considered significant, because implants are rapidly covered by host plasma proteins (Figure 1). The type and amount of plasma proteins adsorbed and retention on the biomaterial’s surface depend on the physical and chemical properties of the surface such as wettability and charge of the material and the characteristics of the plasma proteins such as the relative abundance of plasma, size, stability and charge [44]. The adsorption of plasma proteins, in particular those with adhesive properties such as fibronectin, fibrinogen, vitronectin and von Willebrand factor, on the surface of implanted material provides a physiological microenvironment for the integration of diverse materials into the host tissues. Coating can also concur to modulate the biocompatibility of the biomaterials and influences interfacial events such as blood clotting and thrombogenesis. Beyond such actions, coating with plasma proteins produces a perfect scaffold for the colonization and infection of biomaterials by several species of both Gram-positive (staphylococci) and Gram-negative (for example, *Pseudomonas aeruginosa*) bacteria. For this purpose, bacteria have evolved a series of surface-anchored proteins called MSCRAMMs (microbial surface components recognizing adhesive matrix molecules), each binding to a specific set of host proteins.

Fibronectin (Fn), a multidomain glycoprotein (440 kDa) found in almost all tissues and organs and in biological fluids (plasma concentration, 200 μg/ml), is an important player in processes such as adhesion, differentiation, growth, and migration of cells and is also involved in blood clotting and wound healing [45]. Fn is present in soluble form in various body fluids and in insoluble form in extracellular matrices and basement mem-branes. Hepatocytes produce soluble Fn and secrete it into the bloodstream, whereas the insoluble cellular Fn is synthesized by fibroblasts and endothelial cells. Fn is a dimer made of two nearly identical chains linked by a pair of disulfide bonds. The protein has a modular organization and is composed of a combination of three types of homologous domains, i.e., type I (FnI), II (FnII), and III (FnIII) (Figure 2). Each monomer consists of 12 type I repeats, 2 type II repeats, and 15–17 type III repeats or modules. Cellular Fn can in-clude two variable proportions of alternatively spliced FnIII modules (EIIIB and EIIIA, also termed EDB and EDA, respectively) and one FnIII connecting segment (IIICS). On the other hand, these alternative spliced type III repeats lack in the soluble form. Fn is organized in contiguous domains, each consisting of several modules, and each domain contains binding sites for specific ligands. Five independently folded type I modules (modules FI1–5) constitute the N-terminal domain (NTD) which binds, among others, bacterial receptors and fibrin. The region downstream of the NTD consists of the combination of FnI6FnII1–2FnI7–9 repeats and binds gelatin/collagen. The wide following region including 15 con-stitutive type III repeats comprises a number of modules involved in interactions with other extracellular molecules. Among these, the RGD motif is critical for integrin binding and consequent cell adhesion to the extracellular matrix. The interaction of the RGD motif with integrin α5β1 facilitates the internalization of *S. aureus* by non-professional phagocytic cells. The extreme C-terminal of Fn (FnI10–12) contains a second fibrin binding site and a cysteine residue involved in the interchain disulfide bond (Figure 2) [45,46,47].

Fibrinogen (Fbg) is a large glycoprotein (340 kDa) synthesized in the liver and circulating at a concentration of 2.6 mg/mL. The molecule appears as a trinodular rod and contains three pairs of non-identical chains, Aα (67 kDa), Bβ (56 kDa) and γ (47 kDa). The assembly of Fbg occurs through complex interchain disulfide bonding involving the N-terminal regions of the three polypeptides; together, these form the central nodular E domain. The globular end of the molecule represents the D domain, containing the C-termini of the Bβ and γ chains (Figure 2). Thrombin can cleave an Arg-Gly bond in the N-terminus of the α chain, releasing the 16 residue fibrinopetide A (FPA). The new exposed N-terminal end of the α chain interacts with sites in the γ chain C-termini of neighboring molecules, initiating the non-covalent end-to-end and lateral assembly of the two molecule thick fibrin protofibrils. Subsequently, a second thrombin cleavage releases a 14-residue peptide (FPB) from the N-terminus of the Bβ chain and the exposed new β N- terminus reinforces lateral aggregation resulting in the formation of tick fibrin fibers. Fbg binds specifically to integrin receptors on platelets, endothelial cells and many other cells. Binding to activated platelets is specific and mediated by the platelet integrin αIIβ3. Residues in the γ chain of Fbg from 400 to 411 are necessary for binding to platelets. The adhesion of endothelial cells to surface-coated-Fbg appears to be mediated by α_v_β_3_ integrin and the RGD sequence in the C-terminus Aα chain [48].

Vitronectin (Vn), a glycoprotein synthesized in the liver, is found at a high concentration in plasma (200–700 μg/mL) [49] and is also present in the extracellular matrix of different tissues [50]. The N-terminal portion of mature Vn (43 aa residues) consists of a somatomedin B (SMB) domain followed by the integrin-binding RGD motif. Several members of the integrin family are engaged in Vn binding, among these, integrin α_3_β_1_, α_v_β_3_, α_v_β_5_, and α_5_β_1_ [51]. The next domain comprises four hemopexin-like domains with putative heme-binding motifs. In addition, Vn has three heparin-binding domains corresponding to residues 82–137, 175–219, and 348–361 [52,53,54]. Vn is present in animal organisms as a native, folded monomer in plasma/serum and as a multimeric unfolded form in the extracellular matrix [49,55,56] (Figure 2). Conformational change from the monomeric to the activated, multimeric state is promoted by binding to physiological ligands such as the thrombin-antithrombin complex. Vn conformational activation reveals a number of cryptic sites, including the full exposure of the heparin-binding site at the C-terminal domain of the protein [57,58] and the RGD cell-binding motif [59,60].

Von Willebrand factor (vWF) is a large glycoprotein produced by activated endothelial cells and megakaryocytes, the precursor cells of platelets. vWF circulates in the bloodstream at concentrations between 8–14 μg/mL. The mature 2050 amino acid long monomer consists of three homologous units arranged in the following order: D′D3–A1–A2–A3–D4–C1–C2–C3-C4-C5-C6-CK [61,62] (Figure 2). Each domain can specifically bind to several ligands. The D′D3 domain interacts with factor VIII, the A1 domain binds to the platelet GPIb receptor, heparin, and collagens type IV and VI. The A2 domain contains a cryptic cleavage site for ADAM13 protease, the A3 domain binds fibril-forming collagens I and III and C4 is the binding domain for integrins αIIβ3 and αvβ3 through an RGD motif.

As reported above, plasma protein-coated biomaterials are the specific substrates for a variety of MSCRAMMs expressed by *S. aureus* cells. The MSCRAMM protein family is the most prevalent group of surface proteins that bind to the extracellular matrix and plasma proteins. MSCRAMMs have a conserved overall structure, which includes, from N terminus to C terminus, a signal peptide followed by one or more ligand-binding domains, a cell wall LPXTG anchoring motif, a membrane-spanning region and a short positively charged intracellular tail [63]. The N terminus of several MSCRAMMs contains an A domain made of three separated IgG-like folded subdomains N1, N2 and N3. The combination of N2 and N3 forms a structure that binds the ligand by the “dock, lock and latch” (DLL) mechanism [64,65].

The archetypal examples of staphylococcal MSCRAMMs are clumping factor A (ClfA) [66,67] and B (ClfB) [68,69]. Both the proteins bind Fbg using a variant of the DLL mechanism [66,69]. Linking the A domain to the wall-spanning region W is the region R composed of Ser-Asp repeat (SD repeat) (Figure 3). Other members of the Clf subfamily are SdrC, SdrD and SdrE which contain additional B repeats inserted between the A domain and the SD repeat region [70,71]. Fibronectin-binding protein A (FnBPA) and FnBPB have A domains that are structurally and functionally similar to the A domain of the Clf group and bind several ligands including Fbg [72] and plasminogen [73]. Located in place of the SD repeat region are tandemly repeated motifs (11 in FnBPA and 10 in FnBPB), each binding the NTD of each Fn subunit [74,75] (Figure 3). Importantly, the bridging activity of Fn between FnBPA or FnBPB of *S. aureus* and α_5_β_1_ integrin on the surface of non-phagocytic cells promotes the internalization of *S. aureus* into the host cells [76,77,78].

An important highly expressed MSCRAMM in *S. aureus* is protein A (SpA). The N terminus of SpA comprises a tandem array of five separated folded three-helical bundles, each of which can bind to different ligands (Figure 3). The A1 domain of vWF, tumor necrosis factor receptor 1 and the Fc region of IgG bind to the interface between helices E and D, while the Fab region of IgM binds to the interface of helices D and A [79,80,81,82]. IgG- and vWF-coating of medical device surface can also induce the SpA-mediated attachment of *S. aureus* to implants and qualifies SpA as a genuine adhesin.

A significant number of *S. aureus* strains express CNA, an MSCRAMM that contains a collagen-binding A region at the N-terminus followed by repeated sequences, the B repeats, and a C-terminal cell wall anchoring elements [83] (Figure 3). CNA recognizes collagen by a variation of the DLL mechanism called a collagen hug, a multistep process where the triple helical structure of monomeric collagen is critical for binding [84]. CNA also binds to laminin and C1q and interactions of CNA-laminin and CNA-C1q are weaker than the binding of CNA to collagen [85,86]. CNA binds to the collagenous domain of C1q and interferes with the interaction of C1r with C1q. As a result, C1r2C1s2 is displaced from C1q, and the C1 complex is deactivated [87]. Thus, the dual function of CNA as adhesin and subverter of the classical pathway provides a benefit to pathogen survival and helps establish infection both on the tissues and implants. Interestingly, an epidemic cluster of *S. aureus* from orthopedic periprosthetic infections was identified and characterized from the *bbp-cna* gene tandem for the adhesins binding bone matrix proteins [87]. The cluster corresponded to an epidemic clonal complex, the historical CC30 responsible for worldwide epidemics. The ability to bind bone matrix proteins explains the invasive success of the clone in periprosthetic bone tissue [88].

In iron restricted conditions, *S. aureus* cells express a set of MCSRAMMs named iron-regulated surface determinant (Isd) proteins that have one (IsdA), two (IsdB) or three (IsdH) NEAT (Near iron transporter) domains that bind hemoglobin and heme. Each NEAT motif of IsdB also binds to Vn and vWF, suggesting a role as an iron transporter and adhesin for this protein [89,90] (Figure 3).

The MSCRAMM-mediated adhesion to protein-coated biomaterial is propedeutic to the bacterial colonization and development of biofilm on the surface of medical devices.

### 3.2. Proliferation and Production of Biofilm Matrix

After attachment to the plasma protein-covered device is accomplished, bacteria grow by proliferation and produce polymeric molecules to form a scaffolding extracellular matrix (Figure 1). Biofilm polymers include polysaccharides, proteins, extracellular DNA (eDNA), teichoic acids and other molecules on the whole constituting an aqueous gel-like material with proper mechanical properties and resistance to external shocks [91].

*S. aureus* and *S. epidermidis* produce one main biofilm exopolysaccharide, named polysaccharide intercellular adhesin (PIA). PIA is a β1-6-linked homopolymer of N-acetyl-glycosamine, which has a net positive charge due to the enzyme-catalyzed removal of 15 to 20% of N-acetyl groups after the polymer is secreted. The cationic character of PIA allows electrostatic interaction with other negative charged biofilm components such as teichoic acids. PIA biosynthesis is accomplished by the products of the *ica* locus, which comprises the *icaA*, *icaD*, *icaB* and *icaC* genes [92].

The enzyme IcaA, an N-acetylglucosaminyltransferase, synthesizes PIA from UDP-N-acetylglucosamine precursor. IcaD is required to enhance the efficiency of IcaA while IcaC allows the translocation of PIA across the cell membrane [93]. The bacterial surface-associated enzyme IcaB is responsible for the above-mentioned limited deacetylation of PIA [94]. In addition to the *ica* biosynthetic genes *icaADBC*, the PIA biosynthesis locus contains a negative regulatory gene, *icaR* [95].

Several strains of *S. aureus* and *S. epidermidis* have been reported to not produce PIA but still form biofilm both in vivo and in vitro. In those strains, several staphylococcal surface proteins can promote the accumulation/maturation phase of biofilm development [96,97,98]. The *S aureus* surface protein G (SasG) [99], clumping factor B [100], serine aspartate repeat protein (SdrC) [101], the biofilm-associated protein (Bap) [102,103] and the fibronectin-binding proteins (FnBPA and FnBPB) [104,105] are individually involved in biofilm matrix formation.

SasG, a surface anchored biofilm-forming protein, contains an N-terminal A domain followed by a region composed of a variable number of B repeats. Each B repeat comprises a nearly identical 78 residue G5 sequence and an E region of 50 residues. The mechanism for SasG-mediated cell association in biofilm is based on homophilic protein–protein interactions. Specifically, SasG is enzymatically cleaved within B region to remove the N-terminal A domain and the exposed B repeats on neighboring cells interact with each other in a Zn^2+^-dependent manner, leading to cell–cell adhesion [106].

The A domains of SdrC [101] and FnBPs [105,107] engage in homophylic interactions and promote cell–cell accumulation of staphylococcal cells during biofilm formation. The A domain of SdrC can also promote attachment to abiotic hydrophobic surfaces and could thus be involved in both the early attachment to an indwelling medical device and in accumulation phase of biofilm [108]. Notably, the properties of the PIA- and protein-dependent biofilms show marked morphological differences in structures. The PIA-dependent biofilm is characterized by the presence of fibrous, string- like structures that are absent in proteinaceous biofilm [109,110] and have a higher strength than protein-based biofilm [111].

A number of secreted proteins in the extracellular matrix have been reported to be involved in staphylococcal biofilm formation. Among these we can mention the giant extracellular matrix binding protein (Embp), the extracellular adherence protein (Eap) and the extracellular matrix binding protein (Emp) [14]. The specific molecular mechanism with which these proteins intervene in the process is unknown.

Besides the proteinaceous components, extracellular DNA (eDNA) works in biofilm as an electrostatic net, thus stabilizing and strengthening the matrix [112,113,114,115,116]. Studies on the mechanism of eDNA production and release have revealed that bacteria produce eDNA either through altruistic suicide or fratricide killing and that bacterial autolysis is mediated by murein hydrolase [117]. In the presence of eDNA β-toxin, a sphingomielinase of *S. aureus*, forms covalent cross-links to itself, producing an insoluble nucleoprotein and stabilizing the biofilm structure [118].

### 3.3. Dispersal Phase of Biofilms

In the dispersal phase the biofilm structure is disrupted by enzymatic degradation of the matrix components, most notably by proteases, nucleases and soluble modulins (PSMs) (Figure 1). Among the biofilm-degrading enzymes, proteases are the most important ones [119]. *S. aureus* produces at least ten proteases, each having a proper substrate specificity and expression profile. The major proteases that modulate biofilm integrity include V8, aureolysin and staphopains and their identified targets can be FnBPs and ClfB [100,105,120,121,122]. Cathepsin G released during neutrophil degranulation contributes to the degradation of biofilm into fragments [123] and possibly promotes biofilm dispersal.

The biofilm matrix component eDNA is susceptible to degradation by *S. aureus* nucleases. *S. aureus* is known to produce two secreted nucleases Nuc1 and Nuc2 [124]. The expression of Nuc1 reduces biofilms and a *nuc1* mutation in strain USA300 shows increased in vitro biofilm development [125]. On the other hand, purified Nuc2 prevents biofilm formation in vitro and modestly decreased biomass in dispersal experiments [126].

A group of small α-helical peptides known as phenol soluble modulins (PSMs) functioning as surfactants are also involved in biofilm dispersal. PSMs can be classified in α-type peptides (20–25 amino acids in length) and β-type peptides (45 amino acids in length) [127]. PSMs work on growing biofilms, while they have a limited effect on the dispersal of mature biofilms [128]. They are also involved in the formation of internal channels in biofilms and affect the structure of both PIA-dependent and -independent biofilms [111]. In contrast to these actions, PSMs form amyloids that stabilize *S. aureus* biofilms and block the enzymatic degradation of the biofilm structure [129].

## 4. Regulation of Biofilm Formation

A central regulator controlling biofilm development in staphylococci is a peptide quorum-sensing system, also called the accessory gene regulator or *agr* system. The *agr* locus consists of two divergent transcriptional units, RNAII and RNAIII, whose transcription is driven by the P2 and P3 promoters, respectively. The RNAII locus contains four genes, *agrB*, *agrD*, *agrC* and *agrA*. The *agrD* transcript encodes a peptide precursor of autoinducing peptide (AIP). In its mature form, AIP is 7–9 amino acids long and contains a thiolactone ring between the centrally located cysteine and the C terminus. The *agrB* gene product is a transmembrane endopeptidase that processes AgrD into the final structure with assistance of the type I signal peptidase SspB. Outside the cell, the AIP signal accumulates at particular concentrations level (usually in the low nM range) and activates AgrC. AgrC, the product of the *agrC* gene, is a transmembrane histidine kinase sensor that self-phosphorylates upon binding AIP. Activated AgrC in turn phosphorylates the cognate response regulator AgrA, which then binds to the P2 promoter region for RNAII leading to the autoinduction loop and the P3 promoter region for RNAIII, as well as the promoters controlling the expression of the PSMs. RNAIII is the intracellular effector molecule of the *agr* system responsible for the control of the Agr targets. RNAIII directly inhibits the production of predominantly surface proteins such as protein A and others. Moreover, through the inhibition of the repressor Rot, RNA III leads to the prompt density-dependent upregulation of enterotoxins, α-toxin, leukocidins, degradative exoenzymes such as staphopain proteases, and down-regulation of surface proteins such FnBPs and coagulase. In Staphylococcus, the *agr* system appears to influence biofilm formation at structuring and dispersal stages through its effect on PSM and protease expression. Moreover, the *agr* system controls the expression of surface proteins (adhesins) involved in the initial stage of biofilm development. Notably, the *agr* system does not affect PIA-dependent biofilm formation [130,131,132,133,134].

Besides the *agr* system, biofilm formation is regulated by SarA, a protein which controls *ica* [135,136] and serine protease loci [137]. SarA also indirectly impacts biofilm development through the enhancement of *agr* system activity [138] and expression of nuclease [139]. Sigma B (Sig. B), an alternative sigma factor of RNA polymerase, increases *sar* gene expression and down regulates the activity of the *agr* system [140]. Specifically, the downregulation of RNAIII expression and production of secreted proteases affect the formation of FnBPs-dependent biofilm [141].

## 5. Evasion of the Immune System by Biofilm and Resistance to Antibiotics

Many studies have investigated the effects of staphylococcal biofilms on immune cell function. In the innate immune system, neutrophils and macrophages are the first line of defense against staphylococcal infections. One study reported that neutrophils have a limited influx in a murine model of catheter-associated *S. aureus* biofilm growth [142], while other investigations have demonstrated that activated neutrophils are prevalent at the site of infection in human patients with orthopedic device-associated staphylococcal biofilm infections [143,144]. Moreover, several investigations have reported that human neutrophils in vitro cocultured with *S. aureus* localize to the biofilm and can phagocytose bacteria [145,146]. On the other hand, it has been demonstrated that the *agr* system may facilitate resistance of staphylococcal biofilm to neutrophil killing. Consistent with this, an *agr* mutant resulted in being less cytotoxic to neutrophils in coculture than its wildtype strain, suggesting that bacteria expressing *agr* could kill neutrophils and consequently evade phagocytosis [145].

Along this line, *S. aureus* extracellular nuclease has been reported to degrade NETs (Neutrophil Extracellular Traps), which are secreted thread-like structures composed of DNA and granular antibacterial components [147,148,149].

The anti-biofilm action of macrophages has also been the subject of investigation. Biofilm mode of growth promotes activation of the macrophage phenotype M2 which is characterized by high arginase activity and the production of Il-10, Il-4 and IL-13 [150]. This will create a microenvironment that favors little phagocytosis and even the death of those macrophages that were close to the biofilm [151].

To investigate how *S. aureus* biofilms respond to macrophages and neutrophils gene expression, patterns were profiled using microarrays. The incubation of macrophages with *S. aureus* biofilms led to a global downregulation of genes involved in metabolism, cell wall synthesis/structure and transcription/translation/replication. Unlike coculture with macrophages, coculture *of S. aureus* biofilms with neutrophils did not affect the biofilm transcriptome [152].

The biofilm also increased resistance to antibiotics. This is mainly due to the presence of a matrix in the biofilm that functions as a physical barrier for some antibiotics and to the reduced metabolic processes targeted by antibiotics. Additionally, during prosthetic infection, the *agr* system is downregulated. The consequent absence of PSMs would lead to the pronounced aggregation of bacteria and increased resistance of the infected joint prosthesis and adjacent tissues to the treatment with antibiotics. Hence, novel effective anti-biofilm antibiotics should penetrate the biofilm structure and show high antimicrobial resistance [14,15].

## 6. Therapeutic Strategies to Fight Biomaterial-Associated Staphylococci

Several experimental efforts have been put forth to identify antimicrobials that can treat biomaterial-associated *S. aureus* biofilms. Treatment with DNAase I, which degrades DNA in biofilm, combined with trypsin and proteinase K efficiently disrupts biofilm matrix [122,152,153,154]. Likewise, dispersin B, a glycoside hydrolase, can break down the polysaccharide components of staphylococcal biofilms and can promote penetration of antibiotics [155,156,157]. Hydrolases such as α amylase and cellulase and lysostaphin have also been shown to significantly reduce the matrix biomass of *S. aureus* biofilm [158,159]. However, the in vivo use of exoenzymes as therapeutics may have limited application due to the immunological and inflammatory implications of these treatments.

*cis*-2 decenoic acid (C2DA), a fatty acid produced by *Pseudomonas aeruginosa*, causes a release of planktonic bacteria from *S. aureus* biofilm, suggesting that it can be utilized as a dispersal agent. C2DA combined with traditional antibiotics increases the eradication of *S. aureus* biofilm [160].

Antimicrobial peptides (AMPs), such as a derivative of the cathelicidin peptide LL-37 named OP-145, when incorporated into a medical device, were shown to prevent implant-associated *S. aureus* infections in rabbits [161]. Another LL-37 derivative, SAAP-148, shows to be effective in the eradication of methicillin-resistant *S. aureus* biofilm both in vivo and in vitro [162]. An even better therapeutic efficacy was demonstrated by the combination of AMPs IB-367 and BMAP-28 in the treatment of catheter-associated biofilms [163,164]. As reported for antibiotics, bacteria, including *S. aureus*, have developed specific resistance mechanisms to neutralize the action of AMPs. Thus, caution should be taken when AMP-based treatments are considered.

Other compounds such as aryl rhodamine, D-amino acids, metal chelators and benzimidazole have been successfully tested as anti-biofilm agents in vitro, suggesting the promising use of a variety of small inhibitors of biofilm associated with implanted medical devices [154,165,166,167].

Iron is critical for a variety of cellular processes including energy production, DNA synthesis and biofilm. Analogs of iron have been used to effectively disrupt staphylococcal biofilms: among these, gallium nitrate has been shown to be effective at reducing bacterial biofilm in vitro [168,169]. The metalloporphyrin gallium-protoporphyrin IX (GaPP) is a compound capable of mimicking heme and facilitating metal ion uptake by bacteria. Once inside the bacterial cell, GaPP can block iron-dependent activities of cytochromes, catalases and peroxidases [170]. GaPP administered together with the iron chelator deferiprone demonstrates therapeutical efficacy against biofilm produced by MRSA and MSSA bacteria [171]. Moreover, this combination of compounds strengthened the anti-biofilm activity of antibiotics [172].

Coating with antiadhesive or antibacterial compounds, such as metal ions (copper, silver and zinc) and nanoparticles technology, have also been used as therapeutic strategies to neutralize biomaterial infections [173,174]. However, these approaches do not result in a full clinical success, possibly due to the fact that the medical devices are rapidly covered by plasma proteins so that the anti-biofilm compounds are masked and their efficacy reduced.

Synthetic and natural compound targeting products of the *agr* system have been evaluated for their efficacy against *S. aureus* infections. Among these compounds we can mention ambuic acid, a fungal metabolite that interferes with the biosynthesis of AIP peptide and has been successfully tested in a murine model of skin and soft tissue infection [175]. The plant-derived hamamelitannin and analogues have been shown to potentiate the antibiotic susceptibility of *S. aureus* biofilms by affecting peptidoglycan biosynthesis and eDNA release [176,177,178]. Byaril hydroxyketones, which inhibit the binding of AgrA to P3 promoter, when tested in a *Galleria mellonella* insect larvae infection model, showed increased larval survival. A combination therapy of these quorum-quenching agents with the antibiotics cephalothin or nafcillin revealed additional survival benefits [179]. The antibiofilm value of these blockers in animal bearing implant-associated *S. aureus* infections remains to be determined.

Bacteriophages have also evolved as potential bactericidal agents. The efficacy of phage therapy has been explored for a wide range of *S. aureus* diseases, including biofilm formation on indwelling medical devices. For example, bacteriophage K treatment significantly reduces the *S. aureus* infection of a central venous catheter compared to controls in a rabbit model [180]. Moreover, precoating orthopedic implant surface with phages has been found to prevent *S. aureus* adhesion to medical devices [181] and to reduce bacterial load in adjacent tissues [182]. Finally, orthopedic wires, which are smooth stainless pins used to hold bone fragments together, when coated with phages, were colonized less by *S. aureus* in a murine model of experimental joint infection, and the inhibitory effect could be increased when phage coating was combined with linezolid treatment [182]. In conclusion, although conducted at an experimental level and with animal models, these studies suggest that phage prophylaxis could be an additional tool to prevent *S. aureus* infection of human medical implants.

Vaccination strategies have also been used to contrast the biofilm formation into the medical devices. DNABII proteins are essential components of the biofilm produced by Gram-positive and Gram-negative bacterial species. Estellès et al. produced a monoclonal antibody (mAb) named TLR1068 that cross-reacts with a DNABII epitope and showed that the antibody effectively disrupts the *S. aureus* biofilm in vitro as well as the catheter-associated biofilm in a rat model of infection. Due to this effect, TLR1068 has been proposed as a candidate for the treatment of implant-associated infections in combination with antibiotics [183]. Moreover, immunization with PIA promotes clearance of *S. aureus* from the blood [184] and anti-PIA antibodies are under investigation to treat a variety of biofilm infections in several pathogens [185]. Table 2 refers to the major non-conventional anti-microbial agents used to combat infections by *S. aureus*.

## 7. Concluding Remarks

Progress in current medicine increasingly relies on the use of foreign bodies, but their implantation in many cases creates conditions that favor opportunistic infections by several bacteria, mainly represented by staphylococci. The biofilm mode of growth adopted when *S. aureus* initiates device-associated infections, the persistence of this bacterium in the implanted material due to increased recalcitrance to antibiotics, and lastly, the plethora of immune evasion mechanisms implemented by the bacterium seriously interfere with the success of medical device implantation. Therefore, besides new antibiotics, novel therapeutic approaches need to be pursued, including drugs which disrupt biofilms and the development of anti-biofilm vaccines. The generation of monoclonal antibodies targeting the individual steps of biofilm formation and the single components of biofilm regulatory machinery and their use in passive immunization can broaden the range of therapeutic options.

However, every single intervention to treat patients with infection-associated medical devices is destined to have limited success and for this reason it is desirable to combine and integrate different therapeutic strategies to combat staphylococcal infections. Importantly, a focus on the in vivo evaluation of anti-biofilm therapeutic efficacy deserves better attention. Thus, the use of animal models that reflect conditions closer to the implant-associated infections in humans is highly advisable. Finally, further investigation of the molecular aspects of biofilm development and regulation and a more intense basic and preclinical and clinical research of new compounds and methods will be required.

## Figures and Tables

**Figure 1 ijms-23-05958-f001:**
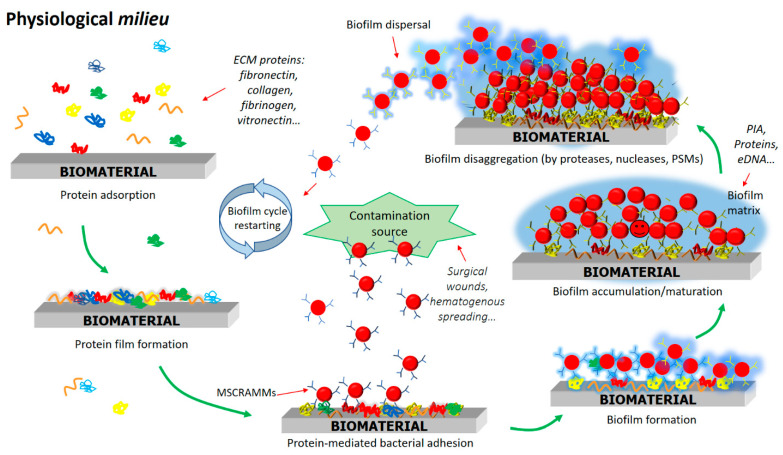
Schematic representation of *S. aureus* biofilm lifecycle. In the first stage *S. aureus* cells directly attach to the abiotic surface of biomaterials, followed by their robust, specific interaction via adhesins (MSCRAMMs) which specifically recognize plasma proteins covering the device. In the second step bacteria proliferate, accumulate and produce a biofilm matrix composed of polysaccharides, DNA and proteins. In the third, dispersal stage several staphylococcal factors disperse the bacteria and disseminate infection. For more details see the text and Refs. [14,15].

**Figure 2 ijms-23-05958-f002:**
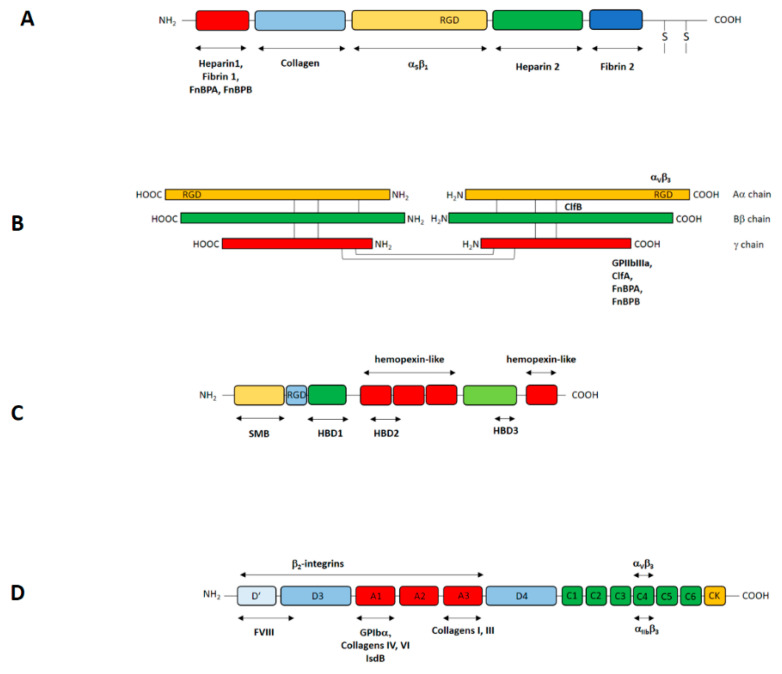
Schematic depiction of the extracellular matrix /plasma human proteins. (**A**) fibronectin; (**B)** fibrinogen; (**C**) vitronectin; (**D**) von Willebrand factor; For each protein, domain organization is indicated. The binding sites for specific ligands on each protein are also reported.

**Figure 3 ijms-23-05958-f003:**
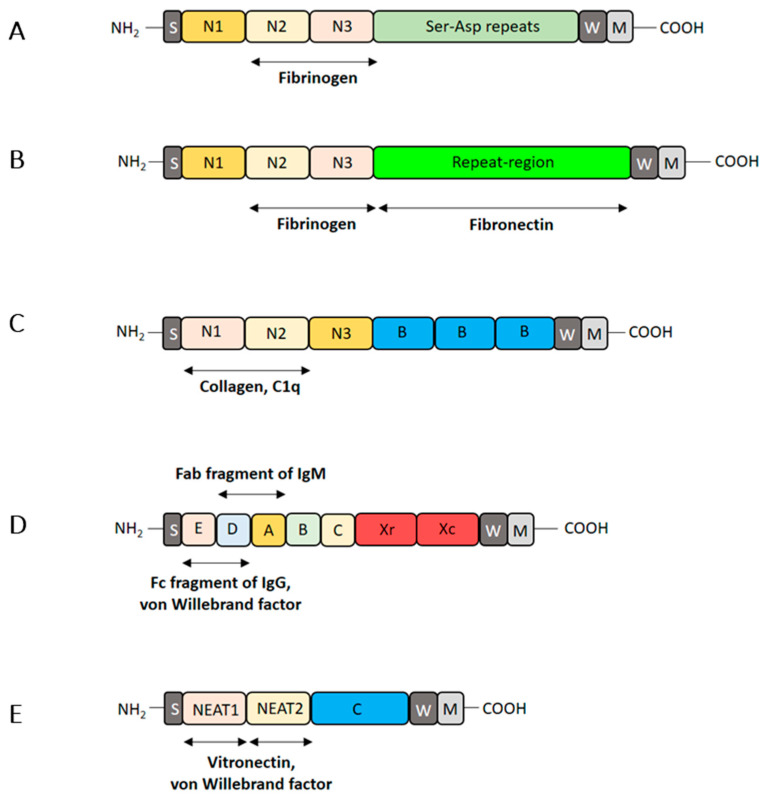
Depiction of *S. aureus* adhesins. (**A**) ClfA (or ClfB); (**B**) FnBPA (FnBPB); (**C**) CNA; (**D**) SpA; (**E**) IsdB. The N-terminal A region of ClfA, ClfB, FnBPA, FnbpB and CNA contains three separate folded subdomains, named N1, N2 and N3. The serine-aspartate region of ClfA and ClfB is composed of a variable number of serine-aspartate dipeptide repeats. The repeat region of FnBPA and FnBPB comprises 11/10 39-residue fibronectin -binding units. The B region of CNA includes a variable number of B repeats. The Xr and Xc of SpA correspond to repetitive and non-repetitive domains, respectively. All the proteins contain a wall binding region (W) and a membrane spanning region (M) at the C-terminus. For each adhesin the binding sites for specific host proteins are indicated.

**Table 1 ijms-23-05958-t001:** Most used devices in human medicine.

Device	Chemical Composition	Application	Type of Infective Agent	Refs
Prosthetic joints	Polyethylene, alumina and zirconia ceramics, silicon carbide, stainless steel, polymethyl methacrylate, titanium	An artificial joint is implanted to replace a damaged or diseased natural joint due to arthritis or other causes such as injures or obesity	*S. aureus*, coagulase-negative staphylococci,	[16,17,18,19,20]
Mechanical heart valves	Titanium and carbon	Artificial heart valves are used to replace heart valves that have become damaged with age or by specific diseases (endocarditis) or congenital abnormalities	*S. epidermidis*, *S. aureus*, streptococci spp.	[21,22,23]
Central venous catheters	Silicone rubber, polyurethane	Central venous catheter is used to give intravenous fluids, blood transfusions, chemotherapy, and other drugs	*S. epidermidis*, *S. aureus*, Enterococci, Aerobic Gram-negative bacilli	[24,25]
Urinary catheters	Silicone, latex	A urinary catheter is used to empty the bladder and collect urine in a drainage bag	*S. epidermidis*, *S. aureus*, *Enterococcus faecalis*, *Escherichia coli*	[26,27,28]
Endotracheal tube	Polyvinyl chloride	Endotracheal tube keepsthe airway open in order to give oxygen, medicine, or anesthesia. It supports breathing in certain illnesses, such as pneumonia, emphysema.	*S. aureus*, *Pseudomonas aeruginosa*	[29,30]
Cerebrospinal shunts	Silicone	Cerebrospinal shunts are used to help drain cerebrospinal fluid and redirect it to another location in the body where it can be reabsorbed	*S. epidermidis*, *S. aureus*, *Propionibacterium acnes*	[31]
Contact lenses	Polymethyl methacrylate/silicone, 2-hydroxyethylmethacrylate/ethylene glycol dimethacrylate	Contact lenses are used to correct nearsightedness, farsightedness, astigmatism and age-related loss of close-up vision, as well as an irregular corneal curvature (keratoconus).	*S. aureus*, *P. aeruginosa*	[32,33,34,35,36]

**Table 2 ijms-23-05958-t002:** List of non-conventional anti-microbial agents used for treating biomaterial-associated *S. aureus* infections.

Antimicrobial Agent	Chemical Composition	Mechanism of Action	Refs
DNAase I/Proteinase K	Enzyme	Disrupt DNA/protein content of biofilm	[122,152,153,154]
Dispersin B	Enzyme	A glycoside hydrolase that can breakdown the polysaccharide component of biofilm	[155,156,157]
*Cis*-2-decanoic acid	Unsaturated fatty acid	Causes release and dispersal of planktonic bacteria from biofilm	[160]
Cathelicidin LL-37 derivatives: OP-145, SAAP-148, IB-367/BMAP-28	Peptide	Disrupt bacterial membrane	[161,162,163,164]
Gallium protoporphyrin IX	A protoporphyrin derivative	Can block iron-dependent activities- of bacterial enzymes	[168,169,170,171,172]
Ambuic acid	A highly functionalized cyclohexenone	Interferes with the biosynthesis of AIP peptide	[175]
Hamamelitannin	2′,5-Digalloylhamamelofuranose	Affects peptidoglycan biosynthesis and DNA release	[176,177,178]
Byaril hydroxyketonecompounds	-	Inhibit the binding of AgrA to P3 promoter	[179]
Phages	Phages are composed of a nucleic acid molecule that is surrounded by a proteinaceous coat	Phage penetrates the cell wall and its DNA is drawn into the bacterium and effectively blocks the bacterium’s ability to function or replicate.	[180,181,182]
TLR1068	Monoclonal antibody	Disrupts the *S. aureus* biofilm in vitro as well as the catheter-associated biofilm in a rat model of infection	[183]

## Data Availability

Not applicable.

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
