# Peer review of "Colonization and Infection of Indwelling Medical Devices by *Staphylococcus aureus* with an Emphasis on Orthopedic Implants"

_ijms, 2022, doi:10.3390/ijms23115958_

Round 1

Reviewer 1 Report

Author Pietrocola et al describe "Colonization and Infection of Indwelling Medical Devices by Staphylococcus aureus with an Emphasis on Orthopedic Implants".

This paper can be accepted after considering the following comments;

1: mention in the abstract this is a review paper

2: First and second paragraphs of the introduction are lacking references.

3: 2. Most Used Medical Devices: Chemical Composition, Applications and Infection>>>>>>will be better if you also make a flow chart (diagram) of different medical devices.

4: Figure 1 will be better to provide some references to the information obtained.

5: Figure 2. Schematic depiction of the extracellular matrix /plasma proteins>>>>>of what human or animals

6:Would be more appropriate if you provide a table of all known molecules used for treating Biomaterial-Associated Staphylococci.

Author Response

Dear Reviewer,

Thank you for taking your time  to revise our manuscript  and for useful comments and suggestions.

1.Mention in the abstract this is a review paper.

We report in the Abstract a sentence  indicating that this work is a review.

2.First and second paragraph of the Introduction  are lacking references-

New references have been incorporated in the text, as suggested.

3. Most used  Medical Devices: chemical composition, applications and Infection... will be better  if you also amake  a flow chart (diagram) of different medical devices.

A new Table (Table1)  referring to the most used medical devices, their chemical composition and potential application in human medicine has been added to the manuscript.

4. Figure1 will be better to provide some references to the information obtained

References  were incorporated in Figure1 Legend, as suggested.

5. Figure 2. Schematic depiction of extracellular matrix/plasma proteins... of what human or animals.

The human nature of the reported host proteins is specified in the Figure Legend.

6. Would be more appropriate  if you provide a Table of all known molecules used for treating Biomaterials asoociated staphylococci.

A table (Table 2) which refers to the major non-conventional, anti-staphylococcal agents used to treat biomaterial-associated  infections has been added to the manuscript.

P.S. Modifications introduced in the text are marked in red.

Reviewer 2 Report

The manuscript makes a good impression because it is well written. In general, the topic of bacterial damage to biomaterials is very relevant. In this regard, I have a comment. In the Introduction, the authors do not comment on how new their review is. It is obvious that such a topical and even "hot" topic could not fail to attract the attention of authors before. Thus, the authors need to justify the novelty of their manuscript.

Author Response

Dear Reviewer,

Thank you for your positive comments and suggestions.

The manuscript makes a good impression because is well written. In general, the topic  of bacterial damage to biomaterilas is very relevant. In this regard, I have a comment. In the Introduction, the authors do not comment on how   new  their review is. It is obvious that such a topical and even "hot" topic could not fail to attract the attention of authors before. Thus, the authors need to justify the novelty of their manuscript.

 The introduction includes a sentence stating the novelty (value) of the work, as suggested.

P.S. Modifications introduced in the text are marked in red.

Round 2

Reviewer 1 Report

Congratulations to all authors

Author Response

We thank the reviewer for taking time  to revise our manuscript  and for useful comments and suggestions and final positive feedback.

Reviewer 2 Report

I continue to wonder about the novelty and usefulness of this review. Not so long ago (2018), the authors have already published a review "Implant infections: adhesion, biofilm formation and immune evasion", and in it they described in sufficient detail the formation of staphylococcus biofilms on implants. I do not see the fundamental novelty of the manuscript in comparison with this review. It cannot be said that the authors updated the data on this issue, since they used the most diverse literature over the years. Thus, the authors need to accurately describe how this manuscript is fundamentally different from the recently published one.

Author Response

We respectfully disagree with the Reviewer about the absence of novelty in our review for the following reasons:

1.In our review we report a section that  refers to  the most  used medical devices, to their chemical composition, and medical applications. To our knowledge, previous reviews on the bacteria/biomaterials issue have not dealt with this argument so analytically.

2. We refer on the biochemical properties of the most important adhesive plasma proteins, (in particular fibronectin, fibrinogen, vitronectin and von Willebrand factor), and their critical role in coating medical devices. As  consequence, adsorption of plasma protein on the implanted material concurs to modulate the biocompatibility of the biomaterials  and produces a scaffold  for the colonization and infection by S. aureus ( and other bacterial species). The molecular aspects of this phenomenon are extensively illustrated in the present work , while they are  marginally mentioned in the recent review published in Nature Rev Microbiol, 2018.  

3. In our review  the biochemical and structural aspects of biofilm development are privileged. For example, we refer to the synergic action of  enzymes involved in PIA formation and to the specific role of surface proteins such FnBPA/FnBPB played in PIA-independent biofilm formation. The mechanisms underlying the action of these biomolecules (polysaccaride and surface proteins) are also reported. Finally, we describe the  molecular aspects of the biofilm regulation and specifically illustrate the biosynthesis and the central regulatory role of AIP in biofilm formation.  On the whole, differently from the Nature Rev Microbiol paper that favors the biological aspects of biofilm development  and medical consequences of this  development, we emphasize the basic biochemical and structural properties  of staphylococcal biofilm.  

4. The review includes a section describing a variety of non-conventional agents used for treating biomaterial- associated infections. This topic is not reported in the Nature Rev Microbiol paper that preferentially highlights the effects of antibiotics and the antibiotic resistance.

5. Both our review and the paper on Nature Rev Microbiol have a section on the “Immunological evasion of the Immune system”. However, while the Nature paper keeps a general profile on the neutralization of the innate immune system by biofilm, in our review we analytically describe the role of neutrophils and macrophages and update their anti-biofilm role on the basis of more recent findings.

 We hope the reviewer finds these arguments/considerations appropriate and compelling and he expresses a final postive evaluation of our review. 

Round 3

Reviewer 2 Report

The authors gave a detailed answer to my doubt about the novelty, highlighting the text in bold so that I would not miss it and which is very similar to screaming or aggression. Nevertheless, if one does not consider emotions, then the authors' arguments look satisfactory. But at the same time, the authors did not make any changes to the text of the manuscript, and in this form it may raise questions not only for me. Usually, authors use the following wording when publishing a series of papers on a topic: "Earlier we considered this issue from one side, and now we are publishing a new paper, where we consider new aspects of this." It is strange that the authors who publish in the journal Nature of Groups do not use these formulations. Also, the authors are inattentive to such a IJMS requirement as "reference numbers should be placed in square brackets [ ]". I hope this will be taken into account in the final layout of the manuscript.

Author Response

Dear Reviewer,

First of all, we would like to point out that it was not our intention to send a message of aggression towards the reviewer. We are sorry for this misunderstanding.

Secondly, we are happy that the reviewer considers satisfactory our arguments concerning the novelty of the review.

In the final part of the introduction, we shortly state that our review is in continuity with a earlier work published in  Nature Rev Microbiol.

Finally, we placed the reference numbers in the text in square brackets, as suggested and  thanks the reviewer for the suggestion.

Round 4

Reviewer 2 Report

The manuscript can be accepted for publication.